# Impact of Torrefaction on Fuel Properties of Aspiration Cleaning Residues

**DOI:** 10.3390/ma15196949

**Published:** 2022-10-07

**Authors:** Barbora Tamelová, Jan Malaťák, Jan Velebil, Arkadiusz Gendek, Monika Aniszewska

**Affiliations:** 1Department of Technological Equipment of Buildings, Faculty of Engineering, Czech University of Life Sciences Prague, Kamýcká 129, 165 21 Prague, Czech Republic; 2Department of Biosystems Engineering, Institute of Mechanical Engineering, Warsaw University of Life Sciences, Nowoursynowska 164, 02-787 Warsaw, Poland

**Keywords:** biochar, torrefaction, calorific potential, fuel properties, stoichiometry, agricultural residues

## Abstract

To maximise the use of biomass for energy purposes, there are various options for converting biomass to biofuels through thermochemical conversion processes, one of which is torrefaction. Higher utilisation of waste from the aspiration cleaning of grains, such as wheat or maize, could be one of the means through which the dependence on fossil fuels could be reduced in the spirit of a circular economy. In this study, the effect of torrefaction on fuel properties of agricultural residues was investigated. The tested materials were waste by-products from the aspiration cleaning of maize grains and waste from wheat. The materials were treated by torrefaction under a nitrogen atmosphere (225 °C, 250 °C, and 275 °C), over a residence time of 30 min. During the treatment, weight loss was monitored as a function of time. Proximate and elemental composition, as well as calorific values, were analysed before and after torrefaction. Torrefaction has a positive effect on the properties of the fuels in the samples studied, as shown by the results. The carbon content increased the most between temperatures of 250 °C and 275 °C, i.e., by 11.7% wt. in waste from maize. The oxygen content in the maize waste samples decreased by 38.99% wt. after torrefaction, and in wheat waste, it decreased by 37.20% wt. compared to the original. The net calorific value increased with increasing temperatures of process and reached a value of 23.56 MJ·kg^−1^ at a peak temperature of 275 °C in by-products from maize. To express the influence of the treatments on combustion behaviour, stoichiometric combustion calculations were performed. Differences of up to 20% in stoichiometric combustion parameters were found between the two types of waste. A similar case was found for fuel consumption, where a difference of 19% was achieved for torrefaction at a temperature of 275 °C, which fundamentally differentiated these fuels.

## 1. Introduction

Although there are a number of available energy sources, biomass represents the world’s fourth largest primary energy source and represents a potential substitute for fossil fuels [1,2]. Nature provides a wide range of biomass resources, and global biomass production is estimated at around 100 billion tonnes per year [3]. Biomass is considered a clean fuel due to its low sulphur and nitrogen content. Less SO_x_ and NO_x_ are produced during combustion and carbon dioxide emissions are approximately zero, effectively reducing the greenhouse effect [1,4]. Biomass waste has become an option for alternative energy production in recent years. [5,6]. Biomass utilisation involves a wide range of potential physicochemical, thermochemical, and biochemical processes [7,8,9,10]. Efficient use of agricultural waste as an energy source is becoming more desirable than ever, due to dwindling fossil fuel supplies and environmental problems caused by high energy consumption [11].

Many agricultural wastes are underused today [12,13,14]. However, there are many thermochemical conversion technologies, such as torrefaction, which have been researched and developed for the treatment of agricultural waste, for example [15]. The resulting products of these technologies can be further converted into various useful biofuels for heat and electricity production [16,17,18,19]. Torrefaction is a thermochemical conversion of biomass and, fundamentally, is a type of pyrolysis process [20,21,22]. Torrefaction is a thermal pre-treatment in which raw biomass is heated to a temperature ranging from 200 °C to 300 °C [23,24,25,26]. It is usually carried out in an inert gas environment, at atmospheric pressure and with a low heating rate, i.e., less than 50 °C min^−1^ [27]. With this process, it is possible to achieve better chemical, physical, and fuel properties of biomass [28,29]. The advantage of torrefaction is the conversion of lignocellulosic raw material into solid biofuel [30,31], which is characterised by higher calorific value, better grindability [32], and higher hydrophobicity [33]. This means that treated biomass does not regain as much moisture during storage. Torrefied fuel has lower H:C and O:C ratios [34], which results in less products of incomplete combustion in flue gas [35], and less water vapour and lower energy losses during combustion or gasification processes. Any biological activity (e.g., rot, mould) is significantly reduced, making torrefied biomass very stable in various storage environments [36]. The biochar can be further processed into compressed fuels. The most common compressed fuels are pellets [37,38], briquettes [39], and granules [40]. Densification technology is an effective and feasible way to improve the properties of biochar [41]. Biochar fuels processed in this way also have the advantage of higher energy density, as well as storage and transport convenience [42,43]. If biochar is used as fuel, it is a carbon-neutral fuel [44]. When it is used for soil treatment, carbon sequestration can be realised and thus negative carbon emissions can be achieved. Biochar and its storage in soil can contribute to a reduction of up to 12% of current anthropogenic CO_2_ emissions [45].

A lot of data on wood biomass torrefaction is currently available [21,46,47,48]. However, less attention has been paid to the torrefaction of agricultural biomass [33]. Maize is the most cultivated crop in the world. In the Czech Republic, it is not ranked first, but it still represents a significant part of the country’s plant production. In 2021, an estimated 770,167 tonnes of maize grain were harvested in the Czech Republic with an average yield of 8.47 t ha^−1^ [49]. Maize grain must be further processed after harvest by cleaning it and drying it. Grain processing on a post-harvest line generates a lot of waste, such as corn spindle residues, stalks, bracts, husks, and broken grains [50]. Wheat is one of the most important grains in the world and is mainly used for food production. The economic and social importance of wheat is due to its widespread production and excellent ability to produce a variety of foods [51]. During wheat cleaning, light fractions, i.e., dust, chaff, and small pieces of grain, are separated and carried by the air stream. According to the Czech Statistical Office, an estimated 4,942,409 tonnes of wheat were harvested in the Czech Republic in 2021 and the average yield was 6.30 t ha^−1^ [49]. One of the opportunities to reduce fossil fuel dependence is the use of all agricultural waste in the circular economy. Therefore, the aim of this research was to assess the fuel properties of selected agricultural waste before and after torrefaction, namely the by-products resulting from the aspiration cleaning of maize and wheat grains.

## 2. Materials and Methods

### 2.1. Samples and Preparation

Samples of waste generated from a grain cleaner were taken from a farm in Central Bohemia, Czech Republic. The samples were taken after a harvest of wheat and maize, at a time when all the material in the dust chamber was sourced from that particular crop. The dust chamber combines light waste fractions from the cleaning of cereals from dust to husks, which are carried by the air stream. In the case of maize, the collected waste contained admixtures of corn cob, bracts, husks, broken grains, and dust. Similarly, the waste from the wheat contained dust, chaff, and grain fragments. Each time, the materials were sampled from the entire cross section of the dust chamber.

Before analyses and torrefaction tests, the samples were left to dry in ambient air to a constant weight and ground to a particle size of under 1 mm by a Retsch SM 100 cutting mill.

### 2.2. Preparation of Samples for Torrefaction

For the preparation of torrefied samples, a programmable weighing furnace, LECO TGA 701, was used. In the furnace, the materials were loaded into crucibles lined with aluminium foil. The furnace was closed, and the crucibles were covered with a lid. First, the samples were dried at 105 °C until they reached a constant weight. Subsequently, nitrogen was introduced, and the furnace was heated to the desired torrefaction temperature, at which it stayed for 30 min. Weight loss was recorded during the whole procedure.

### 2.3. Sample Analysis

All samples were analysed for proximate composition in the LECO TGA 701. The water content was found by drying them at 105 °C until they reached a constant weight, and ash content was found through incineration at 550 °C in an oxygen atmosphere, again until a constant weight. Elemental composition analysis for carbon (*C*), nitrogen (*N*), hydrogen (*H*), and sulphur (*S*) was carried out in the LECO CHN628 + S analyser. The gross calorific value was determined in a LECO AC-600 isoperibol calorimeter. The samples were pressed into pellets and then incinerated. The measurements were repeated at least in triplicates. Net calorific value was determined by calculation based on an elemental and proximate analysis of the individual samples.

Moreover, stoichiometric analyses were determined to calculate combustion characteristics. The results of stoichiometric calculations were transformed to normal gas conditions (temperature t = 0 °C and pressure *p* = 101.325 kPa).

The theoretical amount of oxygen *O*_2,min_ (m^3^ kg^−1^) is based on the following equation:(1)$$O_{2,\min}=V_{m}\left(O_{2}\right)\left(\frac{C}{M\left(C\right)}+\frac{H}{M\left(2\cdot H_{2}\right)}+\frac{S}{M\left(S\right)}-\frac{O}{M\left(O_{2}\right)}\right)$$

*C*, *H*, *S*, and *O* are the contents of carbon, hydrogen, sulphur, and oxygen in the sample (% wt.); *V_m_*(*O*_2_) = 22.39 m^3^ kmol^−1^ is the molar volume of oxygen gas at normal conditions; and *M*(*X*) [kg kmol^−1^] are molar masses of hypothetical species *X* that combine with *O*_2_.

The theoretical amount of dry air *L_min_* (m^3^ kg^−1^) was based on the following equation:(2)$$L_{\min}=O_{2,\min}\cdot \frac{100}{C_{atm}\left(O_{2}\right)}$$
where *C_atm_*(*O*_2_) = 20.95% v. is the volumetric concentration of oxygen in the air.

The theoretical amount of dry flue gas *v_fg,min_* (m^3^ kg^−1^) was based on the equation:(3)$$v_{fg,\min}=\frac{V_{m}\left(CO_{2}\right)}{M(C)}\cdot C+\frac{V_{m}\left(SO_{2}\right)}{M\left(S\right)}\cdot S+\frac{V_{m}\left(N_{2}\right)}{M\left(N_{2}\right)}\cdot N+\frac{C_{atm}\left(N_{2}\right)}{100}\cdot L_{\min}$$
where *V_m_*(*X*) [m^3^ kmol^−1^] are the molar volumes of flue gas components and *C_atm_*(*N*_2_) = 78.05% vol. is the concentration of *N*_2_ in the air.

For each sample, the requested mass flow of fuel to the combustion device was determined according to the required heat output of the combustion plant. The assumed thermal efficiency was 90%, and the rated heat output was varied from 20 kW to 300 kW. The following equation was used to calculate the fuel mass flow rate:(4)$${\dot{m}}_{pv}=\frac{Pk\cdot 100}{qn\cdot \eta }$$
where ${\dot{m}}_{pv}$ is the mass flow of fuel to the combustion chamber (kg·s^−1^), *Pk* is the boiler’s rated heat output (W), *qn* is the fuel net calorific value (J·kg^−1^, J·m^−3^), and *η* is the efficiency of the combustion device (%).

## 3. Results and Discussion

### 3.1. Elemental Analysis and Calorific Value of Waste from Maize

The results of the analyses before and after torrefaction are shown in Table 1. Thanks to the low ash content, it increased only by 1.37% wt. compared to the original material at the highest temperature, 275 °C. In MT-275-30, the ash content was 3.38% wt. Similar numbers have been published: e.g., for corn stalk, the ash content was 2.59% wt. in the original material and 4.43% wt. in the torrefied sample at 260 °C, at the same residence time of 30 min [52].

The carbon content increased during the process at the expense of oxygen and hydrogen. In the torrefied sample at 225 °C, the increase in carbon content is 1.16% wt. higher than in the original. Furthermore, the carbon content increased by 4.14% wt. during the torrefaction process at 250 °C and the largest increase in carbon content (11.7% wt.) occurred at a temperature of 275 °C, reaching 63.03% wt. Overall, the carbon content increased by 33.56% during torrefaction compared to the original sample. In comparison, Zheng et al. [53], who torrefied corn spindles, reported a lower carbon content. Their work states a carbon content of 47.07% wt. at a torrefaction temperature of 245 °C and 50.62% wt. at a temperature of 265 °C. The residence time was 20 min. Medic et al. [54] reported a carbon content of 49.10% wt. in torrefied corn straw at a process temperature of 250 °C and a residence time of 30 min.

The hydrogen content of the sample decreased with increasing torrefaction temperatures. The nitrogen content increased slightly during the process, which may theoretically increase the emission concentrations of nitrogen oxides in the flue gas during combustion [35]. The sulphur content in the sample was negligible.

The oxygen content decreased during torrefaction, in favour of carbon. At 225 °C, the oxygen content was reduced by 1.2% wt. compared to in its original state. Consequently, the oxygen content of the sample was reduced by 3.21% wt. at 250 °C. The largest decrease, 11.7% wt., was recorded at a torrefaction temperature of 275 °C. Thus, the oxygen content decreased by as much as 39% compared to the original sample.

Gross and net calorific values increased with increasing process temperatures. The net calorific value of the original material was 17.51 MJ kg^−1^ in a dry state. The increase in calorific value after torrefaction at 225 °C was not very noticeable; it only increased by 0.36 MJ kg^−1^. With an increasing temperature, the increase in calorific values was more noticeable. The largest increase in calorific value occurred between 250 °C and 275 °C, with a difference of 4.54 MJ kg^−1^ in net calorific value. The net calorific value of the T-275-30 sample was 23.56 MJ kg^−1^. In [52], a net calorific value of 18.72 MJ kg^−1^ in the corn stalk sample was achieved at a torrefaction process temperature of 200 °C, and a calorific value of 21.26 MJ kg^−1^ was achieved at a torrefaction temperature of 260 °C. The residence time was 30 min. Similar results were also obtained by Medic et al. [54], who measured a net calorific value of 19.20 MJ kg^−1^ in a corn straw sample after a torrefaction treatment at 250 °C and a residence time of 30 min. In another piece of research [55], corn stalk was also torrefied, reaching a gross calorific value of 20.13 MJ kg^−1^ after torrefaction at 260 °C and a duration of 30 min, compared to 16.59 MJ kg^−1^ in its original dry state.

### 3.2. Weight Loss of the Waste from Maize during Torrefaction

Figure 1 shows the weight loss curves of maize waste during torrefaction. Weight losses during torrefaction at 225 °C and 250 °C are comparatively mild. During the process at 225 °C, the weight loss was less than 5%. The largest weight loss was recorded at a process temperature of 275 °C, where the weight loss was almost 40%.

### 3.3. Stoichiometric Combustion Analysis of Treated Wastes from Maize

The results of the stoichiometric analysis of maize waste and its biochars are shown in Table 2 and Table 3. The stoichiometric parameters express the requirements of individual materials for combustion air and flue gas production. These combustion air demands increased with the severity of torrefaction treatment. Significant differences in these parameters were determined for a biochar made at 275 °C. However, in energy utilisation, these results are not independent from their calorific values. Therefore, the mass flows of fuel to a combustion device for a given heat output will also change depending on the treatment severity. See Table 3.

### 3.4. Elemental Analysis and Calorific Values of Wheat Waste Samples

The results of the analyses before and after torrefaction are listed in Table 4. Wheat cleaning waste had a relatively high ash content, 16.83% wt. in dry state, which further increased during torrefaction. In sample WT-275-30, the ash content was 24.14% wt. Such a high ash content can be problematic, depending on the type of combustion device, as was observed during the processing of residues from rice cleaning [56]. The reason for the increase in ash content is the loss of mainly moisture but also oxygen and hydrogen during torrefaction.

The carbon content increased with an increasing torrefaction temperature. The largest increase occurred between torrefaction temperatures of 225 °C and 250 °C, where the carbon content increased by 2.36% wt. In sample WT-275-30, the carbon content was 47.56% wt., increasing overall by 13.92%. This is in line with findings by Bai et al. [41], who torrefied wheat straw. Their data show a carbon content of 48.05% wt. in a biochar at a process temperature of 250 °C and a carbon content of 51.30% wt. at a temperature of 275 °C. The duration of stay was also 30 min.

The hydrogen content decreased slightly during the process. The opposite tendency was true for the nitrogen content. The sulphur content in the sample was negligible.

The oxygen content of the sample decreased during torrefaction. During the torrefaction process at 225 °C, it decreased by 3.25% wt. Between process temperatures of 225 °C and 250 °C, the oxygen content was reduced further by almost 5% wt. In the sample WT-250-30, it was 26.25% wt. and in WT-30-275, it was reduced by further 4.62% wt. to 21.62% wt., making a maximum reduction of 37.24% wt. compared to the original material.

Gross and net calorific values increased with process temperatures. The net calorific value of the dry sample was 15.59 MJ kg^−1^. A similar value, 17.80 MJ kg^−1^ in a dry state, was reported for wheat straw by Satpathy et al. [57]. During the process of torrefaction at 225 °C, the calorific value increased by only 0.45 MJ kg^−1^. At a process temperature of 250 °C, the net calorific value increased by 1.31 MJ kg^−1^ against a torrefied sample at 225 °C and, at 275 °C, is increased by a further 0.77 MJ kg^−1^ to the highest value of 18.12 MJ kg^−1^. Cheng et al. [58] reported similar results, where the net calorific value of 19.52 MJ kg^−1^ in torrefied wheat straw was measured at a process temperature of 225 °C and a residence time of 120 min. Additionally, a value of 20.27 MJ kg^−1^ was measured at 250 °C and the same residence time.

### 3.5. Weight Loss of the Waste from Aspiration Cleaning of Wheat during Torrefaction

Figure 2 shows the weight loss of wheat waste during torrefaction as a function of time. The results show that, during torrefaction at 225 °C, there was only a weight loss of 5%. With increasing process temperatures, the total weight loss increased by about 10% wt. with each temperature step; i.e., during torrefaction at 275 °C, the total weight loss was 25%. Similar results were obtained by Cheng et al. [58], when wheat straw was torrefied at set temperatures of 200 °C, 225 °C, and 250 °C.

### 3.6. Stoichiometric Combustion of Sample Waste from Aspiration Cleaning of Grains

The stoichiometric combustion results of the wheat waste and its biochars are shown in Table 5. In contrast to the samples from maize waste, the wheat waste samples show about a ten percent reduction in stoichiometric parameters on dry basis, WT-30-225, and WT-30-250, except for the volume of water vapour. These differences between samples are most pronounced in WT-30-275, where the differences were over twenty percent, again with the exception of volume of water vapour. Similar results were described by Jeníček et al. [59] for torrefied spruce and barley mixtures.

The requested mass flow of fuel was higher by about 19% for samples of waste WT-30-275 compared to the MT-30-275 made from maize waste. However, between WT-30-250 and MT-30-250, there was only about a 3.3% difference. See Table 6. For the original materials, the differences were about 4.4% on a dry basis, and for biochars made at 225 °C, the difference was about 5.0%.

## 4. Conclusions

This study showed that the use of torrefied waste from the aspiration cleaning of grains such as corn and wheat can be an option in reducing the dependence on fossil fuels within a circular economy. Based on the test results presented in this article, the following conclusions have been drawn:

A significant decrease in oxygen content after torrefaction with increasing processing temperatures was confirmed in the tested temperature ranges. In maize waste samples, the oxygen content decreased by 38.99% wt., and in wheat waste, it decreased by 37.20% wt., compared to the original materials.

As the temperature of the torrefaction process increased, the calorific values of all samples increased. This was most pronounced with maize waste, which achieved a calorific value 25.68% higher than that of the original material, after torrefaction at 275 °C and a residence time of 30 min. For the wheat waste sample, the calorific value increased by only 14%.

The unavoidable increase in ash content was limiting the potential for calorific value increase. The highest concentration was over 24% wt. Such a high amount of ash can cause problems for further energy use, e.g., in the operation of a power plant.

During torrefaction, weight loss over time was recorded. This showed a decreasing trend. The highest weight loss, almost 40%, was observed in the maize waste sample after torrefaction at 275 °C.

Stoichiometric analysis of combustion showed significant differences between the original materials and biochars, which could influence the combustion process. Up to 20% of difference in stoichiometric parameters were calculated between the two types of waste. A similar case was found for fuel consumption, where a difference of 19% was achieved for torrefaction temperatures of 275 °C, which fundamentally differentiated these fuels.

The results of the research clearly showed that the properties of the fuel, in all investigated samples, were significantly changed by torrefaction. Torrefaction improved the properties of the fuel compared to the original samples. The analysis showed large concentrations of ash in the samples, which could be limiting for further use as fuel. Therefore, more studies should be carried out to evaluate the quality of the fuels, e.g., an analysis of the properties of residual ash.

## Figures and Tables

**Figure 1 materials-15-06949-f001:**
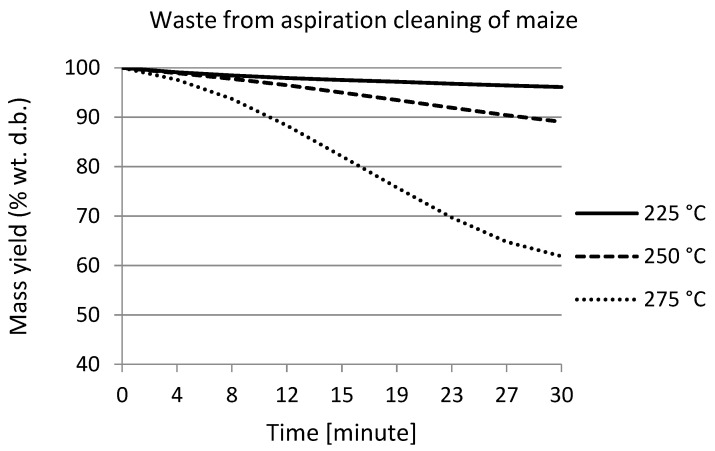
Mass loss curves of maize waste during torrefaction.

**Figure 2 materials-15-06949-f002:**
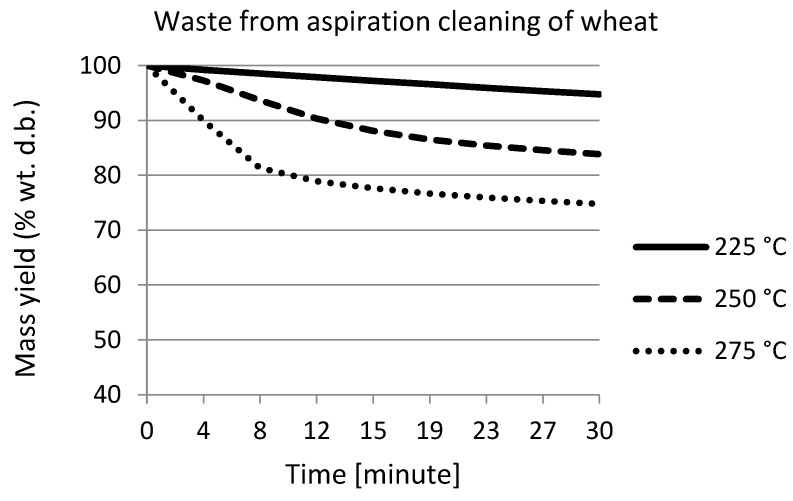
Mass loss curves of wheat waste during torrefaction.

**Table 1 materials-15-06949-t001:** Composition of maize waste before and after torrefaction treatment at varying temperatures and 30 min residence time.

Temp.	C	H	N	S	O	Ash	GCV ^1^	NCV ^2^
°C	% wt.	% wt.	% wt.	% wt.	% wt.	% wt.	MJ·kg^−1^	MJ·kg^−1^
Maize waste	47.20 ± 0.10	6.25 ± 0.08	1.31 ± 0.12	0.11 ± 0.01	43.13	2.01 ± 0.05	18.88 ± 0.10	17.51
MT-30-225	48.36 ± 0.08	6.20 ± 0.07	1.38 ± 0.10	0.09 ± 0.01	41.93	2.04 ± 0.02	19.22 ± 0.09	17.87
MT-30-250	51.34 ± 0.09	6.00 ± 0.07	1.54 ± 0.02	0.08 ± 0.01	38.72	2.32 ± 0.09	20.33 ± 0.09	19.02
MT-30-275	63.03 ± 0.02	5.21 ± 0.08	1.98 ± 0.06	0.09 ± 0.01	26.31	3.38 ± 0.05	24.69 ± 0.10	23.56

Note: ^1^ Gross calorific value. ^2^ Net calorific value. All values were converted to dry state.

**Table 2 materials-15-06949-t002:** Stoichiometric amount of air and specific productions of flue gas components from combustion of sample by-products from maize.

			Dry Basis	T-30-225	T-30-250	T-30-275
L _min_	Stoichiometric volume of air for complete combustion	(m^3^·kg^−1^)	4.41	4.54	4.86	6.10
v_ssp min_	Stoichiometric volume of dry flue gas	(m^3^·kg^−1^)	4.33	4.45	4.76	5.95
v_CO2_	Stoichiometric volume of CO_2_	(m^3^·kg^−1^)	0.88	0.90	0.95	1.17
v_H2O_	Stoichiometric volume of H_2_O	(m^3^·kg^−1^)	0.87	0.87	0.86	0.82
v_N2_	Stoichiometric volume of N_2_	(m^3^·kg^−1^)	3.45	3.55	3.80	4.78
CO_2max_	Concentration of carbon dioxide in dry flue gas after stoichiometric combustion	(% vol.)	20.22	20.15	20.02	19.66

Note: T—torrefaction; 225, 250, 275—temperature (°C); 30—residue time (min).

**Table 3 materials-15-06949-t003:** The mass flow rate of fuel to the combustion device for given heat output of maize waste samples.

Samples		Heat Output (kW)			
		20	50	100	300
Maize waste	Mass flow rate of fuel (kg h^−1^)	4.57	11.42	22.84	68.53
MT-30-225		4.48	11.19	21.38	67.15
MT-30-250		4.21	10.52	21.03	63.09
MT-30-275		3.40	8.49	16.98	50.93

**Table 4 materials-15-06949-t004:** Composition of sample wheat waste before and after torrefaction at varying temperatures and 30 min residence time.

Temp.	C	H	N	S	O	Ash	GCV ^1^	NCV ^2^
°C	% wt.	% wt.	% wt.	% wt.	% wt.	% wt.	MJ·kg^−1^	MJ·kg^−1^
Wheat waste	41.75 ± 0.09	5.20 ± 0.09	1.65 ± 0.12	0.14 ± 0.01	34.43	16.83 ± 0.10	16.72 ± 0.10	15.59
WT-30-225	43.43 ± 0.10	5.12 ± 0.04	1.71 ± 0.10	0.13 ± 0.01	31.18	18.43 ± 0.08	17.16 ± 0.11	16.04
WT-30-250	45.79 ± 0.07	4.73 ± 0.05	1.93 ± 0.07	0.13 ± 0.01	26.25	21.17 ± 0.10	18.38 ± 0.09	17.35
WT-30-275	47.56 ± 0.02	4.42 ± 0.08	2.12 ± 0.01	0.14 ± 0.01	21.62	24.14 ± 0.09	19.09 ± 0.09	18.12

Note: ^1^ Gross calorific value. ^2^ Net calorific value. All values were converted to dry state.

**Table 5 materials-15-06949-t005:** Stoichiometric amount of air and specific productions of flue gas components of wheat waste samples.

			Dry Basis	T-30-225	T-30-250	T-30-275
L _min_	Stoichiometric volume of air for complete combustion	(m^3^·kg^−1^)	3.94	4.17	4.45	4.68
v_ssp min_	Stoichiometric volume of dry flue gas	(m^3^·kg^−1^)	3.86	4.08	4.34	4.55
v_CO2_	Stoichiometric volume of CO_2_	(m^3^·kg^−1^)	0.78	0.81	0.85	0.88
v_H2O_	Stoichiometric volume of H_2_O	(m^3^·kg^−1^)	0.74	0.74	0.70	0.68
v_N2_	Stoichiometric volume of N2	(m^3^·kg^−1^)	3.09	3.27	3.49	3.67
CO_2max_	Concentration of carbon dioxide in dry flue gas after stoichiometric combustion	(% vol.)	20.04	19.75	19.59	19.39

Note: T—torrefaction; 225, 250, 275 temperature (°C); 30 residue time (min).

**Table 6 materials-15-06949-t006:** The mass flow rate of wheat waste and its biochars to a combustion device for a given heat output.

Samples		Heat Output (kW)			
		20	50	100	300
Wheat waste	Mass flow rate of fuel (kg h^−1^)	4.78	11.96	23.92	71.77
WT-30-225		4.66	11.66	23.31	69.93
WT-30-250		4.35	10.88	21.76	65.29
WT-30-275		4.19	10.48	20.95	62.86

## Data Availability

Not applicable.

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
