# Peer review of "Impact of Torrefaction on Fuel Properties of Aspiration Cleaning Residues"

_materials, 2022, doi:10.3390/ma15196949_

Round 1

Reviewer 1 Report

In this study, the effect of torrefaction on fuel properties of agricultural residues was investigated. The tested materials were waste by-products from aspiration cleaning of maize grains and waste from wheat. However, before it could be considered for publication, a major revision is needed as follows:

Point 1: The ABSTRACT section is weak and must be more organized. It suffers from lack of some objectives of this study.

Point 2: Please recheck the manuscript for formatting error and reference style.

Point 4: The importance of this study is not clearly described in the results there must be a DISCUSSION section highlighting novelty of this paper and discussing how much waste byproduct would be required for the proposed solution and what environmental benefits this study would provide. For guidline, please see field implication section of following study and cite them in revised article.

Albeit these papers are not directly related to your study but they could give you a good idea about highlighting the novelty and supplementing discussion pertaining to field implication of the proposed solution. Please write a discussion section and discuss all the aforementioned aspects and cite above articles in the manuscript. This will align the current paper with the aim and scope of the journal. 

Point 5: Please enhance literature review portion by citing latest articles on subject matter and usage of agricultural residues in other applications, please consider citing below articles and other releveant:

https://doi.org/10.1016/j.jclepro.2020.119985

DOI: 10.1016/j.apenergy.2018.10.065

https://doi.org/10.1016/j.jenvman.2020.110285

Point 5: CONCLUSION section have to be rewriten to provide explanation of new contributions and limitation of this study.

Author Response

Point 1: The ABSTRACT section is weak and must be more organized. It suffers from lack of some objectives of this study.

The abstract has been edited.

Point 2: Please recheck the manuscript for formatting error and reference style.

The manuscript has been reviewed.

Point 3: The importance of this study is not clearly described in the results there must be a DISCUSSION section highlighting novelty of this paper and discussing how much waste byproduct would be required for the proposed solution and what environmental benefits this study would provide. For guidline, please see field implication section of following study and cite them in revised article.

Albeit these papers are not directly related to your study but they could give you a good idea about highlighting the novelty and supplementing discussion pertaining to field implication of the proposed solution. Please write a discussion section and discuss all the aforementioned aspects and cite above articles in the manuscript. This will align the current paper with the aim and scope of the journal. 

Thank you for the recommendation. The article is designed so that the results are immediately discussed in one chapter. In the following publications, we will follow the separation of results and discussion chapters.

Point 5: Please enhance literature review portion by citing latest articles on subject matter and usage of agricultural residues in other applications, please consider citing below articles and other releveant:

https://doi.org/10.1016/j.jclepro.2020.119985

DOI: 10.1016/j.apenergy.2018.10.065

https://doi.org/10.1016/j.jenvman.2020.110285

Thanks for the comments, the articles have been cited in this publication.

Point 5: CONCLUSION section have to be rewriten to provide explanation of new contributions and limitation of this study.

We completely rewrote the conclusion.

Reviewer 2 Report

Tamelová and coauthors investigated the effect of torrefaction on fuel properties of waste by-products from aspiration cleaning of maize grains and waste from wheat. Furthermore, elemental component, calorific value, weight loss and stoichiometric combustion were analyzed before and after torrefaction. However, the comparison with previous studies in the ‘Discussion’ section is too general, which cannot support the innovation of research and reflect the importance of research.

Besides, the following questions need to be addressed before publication in Materials.

General comments:

1. I wonder what accounts for these differences on fuel properties of waste by-products from aspiration cleaning of maize grains and waste from wheat.

2. Please elaborate on the significance of high ash content for torrefaction products in Line 207-210.

Other comments:

1. Please supplement the significance tests of the table data.

2. Introduction. The third, fourth and fifth paragraphs should be incorporated into one paragraph.

Author Response

  1. I wonder what accounts for these differences on fuel properties of waste by-products from aspiration cleaning of maize grains and waste from wheat.

Of course, differences can arise due to agro-natural conditions as well as harvesting technology. The differences are due to the harvesting technology, when other impurities can get into the secondary waste. This article was mainly focused on the properties of fuels and thermochemical treatment.

  1. Please elaborate on the significance of high ash content for torrefaction products in Line 207-210.

We elaborate on the significance of high ash content for torrefaction products.

Other comments:

  1. Please supplement the significance tests of the table data.

We have made adjustments.

  1. Introduction. The third, fourth and fifth paragraphs should be incorporated into one paragraph.

We incorporated into one paragraph.

Reviewer 3 Report

The article submitted for review addresses current research issues dt. the conversion of grain cleaning residues into solid biofuels. The discussion lacks an indication in what form and for what systems this type of fuel can be optimally applied. It should be taken into account that the raw material is characterized by a high proportion of fine fractions, which can contribute to significant dusting of the tarified fuel creating a fire hazard. The question is whether agglomeration processes could be one of the options to reduce dust?

Author Response

  1. The article submitted for review addresses current research issues dt. the conversion of grain cleaning residues into solid biofuels. The discussion lacks an indication in what form and for what systems this type of fuel can be optimally applied. It should be taken into account that the raw material is characterized by a high proportion of fine fractions, which can contribute to significant dusting of the tarified fuel creating a fire hazard. The question is whether agglomeration processes could be one of the options to reduce dust?

Thank you very much for your comments. This article was mainly focused on the properties of fuels and thermochemical treatment, the dust problem was not part of the research.

Round 2

Reviewer 1 Report

The authors have addressed all the comments. This paper can be accepted in its current form. I congratulate the authors.

Reviewer 2 Report

The revised manuscript was greatly improved.